# Effects of Food Source Availability, Host Egg:Parasitoid Ratios, and Host Exposure Times on the Developmental Biology of *Megacopta cribraria* Egg Parasitoids

**DOI:** 10.3390/insects14090755

**Published:** 2023-09-09

**Authors:** Sanower Warsi, Ana M. Chicas-Mosier, Rammohan R. Balusu, Alana L. Jacobson, Henry Y. Fadamiro

**Affiliations:** 1Department of Entomology and Plant Pathology, Auburn University, Auburn, AL 36849, USA; szw0132@auburn.edu; 2Center for Environmentally Beneficial Catalysis, University of Kansas, Lawrence, KS 66045, USA; ana.chicasmosier@ku.edu; 3BASF Agricultural Products Group, Research Triangle, Durham, NC 27709, USA; rammohan.balusu@basf.com; 4Department of Entomology, Texas A&M University, College Station, TX 77843, USA; henry.fadamiro@tamu.edu

**Keywords:** ecosystem, Hymenoptera, reproductive attribute, offspring, sex ratio

## Abstract

**Simple Summary:**

This research investigated the reproductive characteristics of two egg parasitoids, *Paratelenomus saccharalis* and *Ooencyrtus nezarae*, which target *Megacoptera cribraria* eggs. We evaluated the impact of different food sources, host egg-to-adult parasitoid ratios, and host exposure times on the number and sex ratio of the parasitoid offspring. The findings showed that a honey solution was the preferred dietary source for *O. nezarae*. Furthermore, a 21:7 host-to-parasitoid ratio yielded the highest and predominantly female offspring for both parasitoids; a host exposure time of three to five days optimized parasitism by *O. nezarae*, whereas a one-day exposure time optimized parasitism by or *P. saccharalis*. These findings contribute to our understanding of the biology of both newly discovered egg parasitoids.

**Abstract:**

Parasitoids forage for hosts in dynamic ecosystems and generally have a short period of time to access hosts. The current study examined the optimal reproductive attributes of two egg parasitoids, *Paratelenomus saccharalis* Dodd (Hymenoptera: Platygastridae) and *Ooencyrtus nezarae* Ishii (Hymenoptera: Encyrtidae), of the kudzu bug, *Megacopta cribraria* Fabricius (Hemiptera: Plataspidae). The proportion of *O. nezarae* and *P. saccharalis* adult offspring that emerged from *M. cribraria* eggs and the sex ratio of the parasitoid offspring were compared among treatments for the effects of different adult parasitoid food sources, host egg-to-adult parasitoid ratios, and host exposure times. Our results suggest that honey solution as a food source, a 21:7 host-to-parasitoid ratio, and three-to-five days of exposure time optimized the production of female *O. nezarae offspring*. For *P. saccharalis*, honey solution as a food source, a 21:7 host-to-parasitoid ratio, and one day were optimal for maximizing female offspring production. These findings provide new information about the biology of these egg parasitoids.

## 1. Introduction

The kudzu bug, *Megacopta cribraria* Fabricius (Hemiptera: Plastaspidae), is an invasive pest of soybeans, *Glycine max* Linneanus, accidentally introduced from Asia to the Southern United States [1]. At present, synthetic chemical insecticides are the primary control method for *M. cribraria* [2]. Repeated interventions of insecticides may cause insect resistance and eliminate beneficial species [3]. Parasitoids may be an ecologically benign, efficient, and cost-effective alternative to regulate the *M. cribraria* population, so preserving parasitoid species richness and activity is crucial [4,5]. In the past decade, two egg parasitoids of *M. cribraria*, *Paratelenomus saccharalis* Dodd (Hymenoptera: Platygastridae) and *Ooencyrtus nezarae* Ishii (Hymenoptera: Encyrtidae), were discovered in the United States. Although parasitism of *M. cribraria* by *O. nezarae* and *P. saccharalis* is known [6,7], there are gaps in knowledge about the biology of both parasitoids in the United States. Given the ubiquity of parasitoids and their importance in influencing the abundance and dynamics of their hosts [8], it is essential to investigate the factors that affect their performance. This study aimed to investigate the effect of food sources, host-to-parasitoid ratio, and exposure time on the parasitism and reproductive efficacy of both parasitoids.

Several studies have shown that the presence of adult food sources maximizes the reproductive success of parasitoids by influencing emergence, female: male ratio, and longevity [9,10,11,12]. A carbohydrate-rich diet in animals can lead to a rise in glucose content in the follicular fluid, which increases the number and size of ovarian follicles [13]. Ingesting sugar may reduce the rate of lipid decline, thereby potentially saving reserves for oocyte maturation [14] and stored sugar as trehalose or glycogen, which may be used in embryogenesis later [15]. Some studies suggested that diets rich in carbohydrates may decrease the mortality risk and increase the residual lifespan of insects by inhibiting inflammatory pathways, thereby decreasing biological aging and oxidative stress [16]. Female parasitoid aging can affect the offspring sex ratio as sperm are generally depleted, or the viability of sperm is reduced over time, which can increase the production of males [17,18,19,20]. Control of sperm release from the spermatheca is generally weakened with wasp age due to physiological or physical defects in the maternal reproductive organs (or senescence) [18].

Host-to-parasitoid ratios and host exposure times can also alter the parasitoid offspring’s fitness by influencing the series of host selection decisions made by adult parasitoids before oviposition [21]. In general, female wasps should select a high-quality host because it provides sufficient food resources for developing embryos, increasing survivability [22,23]. However, when high-quality hosts are limited, a parasitoid can accept a host that has already been parasitized by conspecific females [24], even though sharing resources with competitors can decrease survival [25]. When the parasitoids accept poor-quality hosts, they may allocate fewer resources to male offspring production, which can further bias the offspring sex ratio toward females [26].

This study on the availability of food sources, host/parasitoid ratio, and host exposure times was driven by three different goals: (1) to investigate the factors affecting host selection, reproduction, and offspring survival; (2) to better understand how the absence or presence of food alters offspring production and sex ratios; and (3) to examine how changes in parasitoid to host ratios alter offspring survival, development, and fitness. This information will deepen our understanding of host–parasitoid interactions, factors that optimize reproductive efficacy, and their potential implications for ecological systems. The following hypotheses were tested: (i) the presence of food would enhance both parasitoid species’ emergence and longevity; (ii) wasps would produce a male-biased sex ratio later in their lifespan; (iii) wasp emergence would be lower when the host-to-parasitoid ratio is low or with longer exposure periods; and (iv) parasitoid offspring would be male-biased when female wasps are host-limited.

## 2. Materials and Methods

### 2.1. Plant Material and Growth Conditions

Soybean seeds (var. Pioneer P49T97R-SA2P) were sown in 15.24 cm diameter pots with a depth of 14.22 cm, using Sunshine #8 potting mixture (SunGro Horticulture, Agawam, MA, USA). They were grown in a pest-free incubator at 26 ± 2 °C and 55 ± 5% RH [24]. Plants were watered daily (~200 mL per pot) and fertilized (~15 gm per 3.78 L) according to the label (Scotts-Sierra Horticultural Product Company, Marysville, OH, USA) once a week until use for *M. cribraria* rearing.

### 2.2. Insect Colony

From May to October 2022, adult *M. cribraria* were collected from kudzu in Auburn, AL, and reared in ventilated 30 cm × 30 cm × 30 cm plastic cages (BugDorm-2, Megaview Science Education Services Co., Ltd., Taichung, Taiwan) at 28 ± 5 °C, 16:8 h (L:D), and 50% RH in an environmental chamber (Percival, Perry, IA, USA) and provided organic green bean, *Phaseolus vulgaris* L., pods and vegetative-stage (V2–V3) plants of soybean, *Glycine max* L. New soybean plants were provided weekly [27]. Cages were checked daily for fresh, milky-white eggs (≤24 h), distinguishing them from the darker-colored, aged eggs (>24 h).

The parasitoid species used in the experiments originated from a soybean field in Auburn, AL. Soybean leaves with parasitized (grey color) *M. cribraria* egg masses were collected from the field. Each leaf was trimmed to keep only the sections containing the egg masses and was then placed into an individual 59.1 mL (top diameter: 6 cm, bottom diameter: 4.4 cm, and height: 2.8 cm) condiment cup with approximately 20 pin holes made on the rearing cup wall for aeration and to prevent condensation. All parasitoid colonies and experiments were conducted in these containers and incubated at 25 ± 1 °C, 14:10 (L:D) h, and 75 ± 5% RH photoperiod to maximize possible emergence [27,28]. *Ooencyrtus* nezarae and *P. saccharalis* were distinguished following the characteristics provided by Gardner et al. [6] and Ademokoya et al. [7], respectively, and were separated into different rearing cups. Colonies of the wasps were reared by providing adults with a honey solution (70% honey to 30% water, *v*/*v*) and allowing them to oviposit into ≤24 h old *M. cribraria* eggs.

#### 2.2.1. Experimental Designs Effect of Sugar Feeding on the Reproductive Fitness and Longevity of *P. saccharalis* and *O. nezarae*

Newly emerged (<24 h) adult males and females of *P. saccharalis* or *O. nezarae* were allowed to mate for 24 h. Individuals had no access to honey, water, or hosts during the period from emergence to mating. On day two, wasps were individually placed into 59.1 mL condiment cups (dimension provided above). Nine adult food treatments were included in the experiments following a completely randomized design (CRD). Individual adults of each wasp species were offered (i) water + hosts, (ii) honey + hosts, (iii) water + honey + hosts, (iv) no water, no honey + hosts, (v) water, (vi) honey, (vii) water + honey, and (viii) fasting (no water, no honey). A control group was included consisting of host eggs that were not exposed to parasitoids to assess natural host mortality. Treatments i–iv (i.e., host presence) were only tested for females since males of both wasp species did not exploit the hosts. Treatments v–viii (i.e., host absence) were tested for both males and females separately. A total of 20 replicates of each treatment were conducted for each parasitoid species. Each treatment lasted for the entire life of the parasitoids.

Water was provided from a cotton string extending from a hole made at the bottom of a 0.5 mL Eppendorf microcentrifuge tube (Fisher Scientific, Waltham, MA, USA) filled with de-ionized water in the treatments (i, iii, v, and vii). To prevent insect desiccation while maintaining the distinction between water-provided treatments (i, iii, v, and vii) and water-deprived treatments (ii, iv, vi, and viii), we controlled humidity across the treatments. In the water-provided treatments, insects had direct access to a water source. In contrast, for the water-deprived treatments, we placed a water source in the setup to maintain ambient humidity, but insects were physically prevented from accessing this water for consumption. This design allowed us to mitigate the risk of desiccation in water-deprived treatments by ensuring a stable level of humidity while still preventing the insects from consuming water. The 70% honey solution was kept in a 0.5 mL Eppendorf microcentrifuge tube with a hole at the bottom. A cotton string was threaded through the hole in honey provision treatments (ii, iii, vi, and viii).

Each parasitoid was exposed to 24–30 host eggs that were less than 24 h old (treatments i–iv). The eggs were replaced with fresh host eggs after 24 h, and parasitoid survivability was recorded. The parasitoid-exposed host eggs were transferred to a new cup and maintained in the incubator. Each cup was monitored for 14–16 days until the emergence of wasps. The number of parasitized eggs, the emerged *M. cribraria* nymphs, emerged wasps, and their sex ratio (female/total emerged wasps) in treatments (i–iv) with host eggs were recorded. The survivability of adult individuals was monitored daily in treatments without host eggs (v–ix).

#### 2.2.2. Effect of Host Eggs: Parasitoid Ratios and Host Exposure Times on the Fitness of *P. saccharalis* and *O. nezarae*

In this objective, two experiments were conducted with adult female parasitoids of each species that were mated and naïve (no prior oviposition experience). In one set of experiments, data were collected on an individual female wasp. In the second experiment, data were collected on seven female wasps released together. For both sets of experiments, combinations of two factors were studied and compared; wasps were released into experimental arenas with host egg densities of 21, 42, 84, or 168 (±7) and exposure times of 1, 3, or 5 days. Thus, the host-to-parasitoid ratio for the single and group wasp experiments were 21:1, 42:1, 84:1, or 168:1 and 21:7, 42:7, 84:7, or 168:7, respectively. A CRD was used, and each host density x host exposure time was replicated 20 times for *O. nezarae* and 10 times for *P. saccharalis* due to the availability of parasitoids. We selected the maximum host density of 168 based on a preliminary experiment in which we observed that, on average, a wasp parasitized one egg per hour with a high host density of 150 eggs and over a period ranging from 6 to 24 h. Based on this, seven wasps parasitizing one egg/hour for 24 h could parasitize 168 eggs with little to no superparasitism. The exposure times were chosen based on a preliminary experiment that showed *M. cribraria* nymphs start to emerge on day seven. In all treatments, honey solution (70% *v*/*v*) was provided to the wasp throughout the experiment, as described above.

After the host exposure period ended, parasitized eggs were held for 14–17 days to allow time for *O. nezarae* and *P. saccharalis* adults to emerge [29,30] (personal observation). After the holding period, the final proportion of parasitized *M. cribraria* eggs to emerged *M. cribraria* nymphs, the number of emerged parasitoids, the sex ratio of emerged parasitoids, and the proportion of unascribed eggs (host eggs did not result in emergence) to the total number of eggs per egg mass were recorded.

### 2.3. Statistical Analysis

In the experiment examining the effects of food on adult survival, the Kaplan–Meier method was used. GraphPad Prism version 8.4.2 for Windows 10 (GraphPad Software, La Jolla, CA, USA) was used for survival analysis and generating survival curves. The log-rank (Mantel–Cox) test was used to compare the survival curves (Figure 1, Figure 2 and Figure 3) after confirming that they followed a normal distribution (Shapiro–Wilk, Cramer–von Mises, Anderson–Darling, and Kolmogorov–Sminorv’s tests) and homoscedasticity (Barllet’s, Brown–Forsythe’s, and Lavene’s tests). Two-way ANOVA was used in SAS version 9.4. (SAS Institute, Cary, NC, USA) to determine the effect of food source and wasp age treatments on the proportion of parasitized host eggs (parasitized eggs/eggs in the patch; Appendix A), the proportion of emerged *M. cribraria* nymphs (nymphs/eggs in the patch; Appendix A), the proportion of wasp offspring (emerged wasps/eggs in the patch; Appendix A), and the proportion of offspring sex ratio (female offspring/emerged wasps; Appendix A). GraphPad Prism software was used to generate graphs for parasitized host eggs (Figure 2A and Figure 4A), emerged *M. cribraria* nymphs (Figure 2B and Figure 4B), wasp offspring (Figure 2C and Figure 4C), and offspring sex ratio (Figure 2D and Figure 4D). We also determined the effect of food sources on the average ± S.E. lifetime offspring production (sum of total emerged wasps per female; Table 1) and sex ratio (sum of females/sum of total emerged wasps per female; Table 2) of the parasitoids by conducting one-way ANOVAs followed by Tukey’s test (α = 0.05) using SAS version 9.4.

The effects of host-to-parasitoid ratio and host exposure time factors on the parasitism and reproductive efficacy of individual wasps and groups of seven wasps were analyzed as the proportion of parasitized host eggs (Figure 5A, Figure 6A, Figure 7A and Figure 8A; parasitized eggs/total eggs in the patch), the proportion of emerged *M. cribraria* nymphs (Figure 5B, Figure 6B, Figure 7B and Figure 8B; nymphs hatched/total eggs in the patch), the proportion of wasp offspring (Figure 5C, Figure 6C, Figure 7C and Figure 8C; emerged wasps/total eggs in the patch), and the proportion of wasp offspring sex ratio (Figure 5D, Figure 6D, Figure 7D and Figure 8D; females/total emerged wasps). These figures display averages and standard errors. The data were transformed and tested for normality in preliminary analyses conducted with SAS version 9.4 (SAS Institute, Cary, NC, USA). The results showed that the proportion data from the host-to-parasitoid ratio and the host exposure times experiment were not normally distributed. Different host-to-parasitoid ratio and host exposure times treatment groups had unequal variances (Barllet’s, Brown–Forsythe’s, and Lavene’s tests). Therefore, nonparametric statistical analyses were performed using Software JASP version 0.16.4 (University of Amsterdam, Netherlands) to conduct tests and generate graphs. A Kruskal–Wallis test, followed by Dunn’s multiple comparisons tests, was used to test the effect of more than two treatments (i.e., 21, 42, 84, or 168 host densities and 1, 3, or 5 days of host exposure times). For these statistical tests, the significance level (*p* ≤ 0.05) was adjusted with the Bonferroni method by dividing the significance level by the number of comparisons being conducted. Statistics and significance values of Kruskal–Wallis and Dunn’s multiple comparisons tests are displayed in Appendix A.

## 3. Results

### 3.1. Effect of Food Availability on the Life History Traits of O. nezarae and P. saccharalis

#### 3.1.1. Life History Traits of *O. nezarae* in the Presence or Absence of Food Sources

Survivorship curves for both sexes of *O. nezarae* are shown in Figure 1. Table 1 displays the average lifespan of *O. nezarae* adults for each combination of diet, sex, and host. When *O. nezarae* females were provided with different diet regimens and placed in the presence of an oviposition host (Females + host eggs), starved or water-provided females generally exhibited longer longevity compared to females without host eggs (Figure 1A). This result suggests that even in the presence of food, female longevity is reduced following oviposition. Females with access to honey, regardless of the availability of water, lived longer than their honey-deprived counterparts in the presence or absence of the host (Table 1). Males also had the longest average lifespan when kept with honey, independent of water (Table 1), indicating that honey (or sugar source) maximizes the longevity of *O. nezarae* individuals (Figure 1B). However, female longevity was around two times higher than that of males in all comparative treatments.

*Ooencyrtus nezarae* parasitism of *M. cribraria* eggs was significantly influenced by the diet and age of the parasitoid (see Appendix A for statistics). Honey provisions increased the proportion and duration of parasitism by *O. nezarae* females compared to honey-deprived females, independent of water. Wasps with access to honey were able to parasitize up to 128 host eggs throughout their lifetime, nearly quadrupling the parasitism of wasps that did not have access to honey. Notably, parasitism reached its peak relatively early in the post-emergence phase (Figure 2A). Parasitism consistently declined with age, and this trend was observed across all diet types. The decline was more pronounced for those on water and starved diets than other dietary treatments (Figure 2A). The ability of *O. nezarae* to parasitize the host declined sooner for those on fasting and water diets compared to those on honey and honey + water diets. At these specific ages, the probability of finding a living wasp was 20% for fasting, 10% for water, 30% for honey, and 20% for honey + water (Figure 1A).

In the control, wasp absence resulted in a significantly greater proportion of emerged *M. cribraria* nymphs than wasp presence (F = 15.82, df = 1, 339, *p* < 0.0001). In the absence of a parasitoid, the average proportion of nymph eclosion was 0.86 ± 0.04. The presence of parasitoids influenced the percentage of hatched host eggs; in starved and water treatments, approximately 75% of host eggs hatched into nymphs. The age of female wasps also factored into the proportion of successful *M. cribraria* nymph eclosion (Appendix A). More than 80% of the host eggs yielded nymphs when wasps were greater than 18 days old, regardless of diet.

Diet and parasitoid age significantly influenced the proportion of parasitoid emergence (Appendix A). Females with access to honey, independent of water, produced more offspring than honey-deprived females (Table 2). There was a difference in the timing of reproductive peaks based on the diet regimes of *O. nezarae* females. Females fed with honey, alone or with water, had delayed reproductive peaks. Conversely, those given only water or starved exhibited earlier reproductive surges, possibly as a strategy to ensure offspring production before energy depletion (Figure 2C). The proportion of offspring emergence decreased with the increasing age of the females, showing a clear effect of age on the emergence and viability of eggs.

The offspring sex ratio (female/total emerged wasps) was affected by diet and age (Table 2). There was a reduction in the proportion of female offspring, independent of diet, suggesting that sperm depletion occurred over time (Figure 2D). This effect is more clearly observed for the honey + water-fed wasps; the proportion of female offspring fell below 0.33 after 13 days (Figure 2D). Throughout the entire adulthood of *O. nezarae*, the sex ratio favored males in honey-deprived wasps compared to honey-fed wasps (Table 2).

#### 3.1.2. Life History Traits of *P. saccharalis* in the Presence or Absence of Food Sources

Survivorship curves for both sexes of *P. saccharalis* are shown in Figure 3. Host presence resulted in significantly lower survival of *P. saccharalis* females compared to host absence. Table 1 displays the average lifespan of *P. saccharalis* adults for each combination of diet, sex, and host. Both in host presence and absence, females with access to honey, independent of water, had greater longevity than honey-deprived females (Table 1). Similar results were obtained for *P. saccharalis* males (Figure 3B). Honey-provisioned males also lived significantly longer than honey-deprived males (Table 1). Honey increased average longevity by approximately five times for females (without a host) and males compared with individuals who did not have access to honey. There were no significant differences in the longevity of males and females without a host.

Although the diet did not show a statistically significant impact on the host parasitism of *P. saccharalis* (Appendix A), there was a noticeable trend. Females provided with honey or honey + water showed numerically higher host parasitism than those starved or only given water (Figure 4A). Honey-deprived wasps parasitized 11–43 host eggs throughout their lifespan, whereas honey-provisioned wasps were able to parasitize up to 90 eggs throughout their lifespan. The proportion of parasitized host eggs by *P. saccharalis* consistently declined as female wasps aged. A sharp decline in parasitism was observed in starved and water-provisioned wasps right before death. The ability of *P. saccharalis* to parasitize was entirely depleted at approximately three days post-emergence when fasting, four days with water, 14 days with honey, and nine days with both honey and water. The probability of finding a live wasp at each age was 62.5% for fasting, 25% for water, 25% for honey, and 37.5% for honey + water (Figure 3A).

A significantly lower proportion of emerged *M. cribraria* nymphs were observed when *M. cribraria* eggs were exposed to a female wasp than the control (F = 31.75, df = 1, 193, *p* < 0.0001). The proportion of emerged *M. cribraria* nymphs was nearly double in the absence of a parasitoid (Figure 4B). Diet availability also influenced the proportion of *M. cribraria* nymphs, and more host nymphs survived when female parasitoids were starved (Figure 4B). Female age significantly influenced the proportion of emerged *M. cribraria* nymphs (Appendix A). More than half of the host eggs yielded nymphs when exposed to female wasps over 3 days old, regardless of the diet.

The proportion of *P. saccharalis* offspring was not affected by diet (Appendix A). Honey-provisioned females had higher offspring production throughout their lifespan than honey-deprived females (Figure 4C and Table 2). Females produced the highest average offspring in the first two days after adult emergence, independent of diet (Figure 4C). The proportion of offspring production in females declined as their age increased.

The offspring sex ratio (female/total emerged wasps) was affected by the parasitoid’s diet (Appendix A). However, the average lifetime offspring sex ratio did not differ between starved and fed females (Table 2). Female offspring production was predominant early in the reproductive phase, regardless of diet. However, as the maternal age increased, there was a noticeable shift toward a male-biased sex ratio (Figure 4D).

### 3.2. Effect of Host Egg: Parasitoid Ratios and Host Exposure Times on the Biology of O. nezarae and P. saccharalis

#### 3.2.1. Biology of *O. nezarae* under Different Combinations of Host Egg: Parasitoid Ratios and Host Exposure Times

Host parasitism was influenced by host density and exposure time by parasitoids either foraging singly or in a group of seven individuals (for statistics, see Appendix A). When a wasp was foraging singly, the highest proportion of parasitized eggs (~0.70) was obtained at the lowest host density (21 eggs) over the higher exposure times (three-to-five days) (Figure 5A). However, at the higher host densities (48, 84, and 168), the proportion of parasitized eggs was less than 0.40 at all exposure times. When the number of wasps increased from one to seven, the proportion of parasitized eggs was consistently greater than 0.65 at higher host egg exposure times (three-to-five days) at each host density (Figure 6A). Similarly, the proportion of hatched nymphs was significantly affected by host density (Appendix A) and exposure time (Appendix A). The release of multiple wasps (Figure 6B) significantly reduced the eclosion of *M. cribraria* nymphs compared to a single-wasp release (Figure 5B).

Wasp emergence was also influenced by host density in both single and multiple-wasp release experiments (Appendix A). Host exposure time did not affect the proportion of emerged wasps in the multiple-wasp release experiments but affected the proportion that emerged in the single-wasp experiment (Appendix A). In the single-wasp release experiment, the proportion of emerged parasitoids increased as the host exposure time increased from one to five days at different host densities. A consistent declining trend of wasp emergence was observed as the host density increased (Figure 5C). The proportion of emerged wasps was up to 0.63 for 21 eggs, 0.50 for 48 eggs, and 0.23 for 84 and 168 eggs (Figure 5C). An overall 79% decline in parasitoid emergence was observed as host density increased in the single-wasp release experiment. However, when wasps were released in a group of seven, the proportion of emerged wasps decreased as the host exposure time increased at low host densities (21 and 48 eggs; Figure 6C), opposite to the single-wasp release experiment. In the treatment with a host density of 21, the proportion of wasps fell from 0.75 to 0.27 as the host exposure time increased from one to five days (Figure 6C), indicating that there might be competition among conspecifics for limited host resources. The sex ratio of parasitoid offspring was almost equal (1:1) in all treatments in the single (Figure 5D) and multiple (Figure 6D) wasp experiments.

#### 3.2.2. Biology of *P. saccharalis* under Different Combinations of Host Eggs/Parasitoid Ratios and Host Exposure Times

Host parasitism was influenced by host density, not exposure times, regardless of whether wasps were released individually or in groups of seven (Appendix A). It indicates that the number of hosts is more important than the time available for parasitism. At each host exposure time and in both experiments, a lower proportion of *M. cribraria* eggs were parasitized as the host density increased. In the single-wasp release experiment, the proportion of parasitized eggs was more than 0.70 at lower host densities (21 and 48 eggs) over higher exposure times (three and five days) (Figure 7A). However, in multiple-wasp releases, the proportion of parasitized eggs was 0.99 in most treatments (Figure 8A). More emerged nymphs were observed in the single-wasp experiment (Figure 7B) compared to the multiple-wasp experiment (Figure 8B). The proportion of emerged nymphs in the single-wasp release test was noticeably higher at the density of 168 eggs at each host exposure time (Figure 8B).

Similarly, only host density affected the proportion of emerged wasps in either experiment (single or multiple-wasp release, Appendix A). As the host density increased, the proportion of emerged wasps decreased at each exposure time when a single wasp was exposed to *M. cribraria* eggs (Figure 7C). However, when multiple wasps were exposed, the results varied with the host density and exposure time (Figure 8C). The highest number of wasps emerged after three days of exposure at each host density (Figure 8C). The sex ratio of offspring was also affected by host density only (Appendix A), not by host exposure time (Appendix A). The proportion of parasitoid female offspring was not lower than 0.50 in most treatments, whether host eggs were exposed to a single wasp (Figure 7D) or multiple wasps (Figure 8D).

## 4. Discussion

### 4.1. Effect of Food Availability on the Life History Traits of O. nezarae and P. saccharalis

#### 4.1.1. Survival Curves

The lifespan of parasitoids can be determined by sex [31], particularly when males and females allocate nutritional resources differently for survival and reproduction [32]. In this study, the impact of honey with/without a water supply on adult *O. nezarae* and *P. saccharalis* was similar between the sexes. When provided with honey and water, *O. nezarae* and *P. saccharalis* lived 28–47 (Figure 1) and 28 days (Figure 3), respectively. When provided with honey alone, *O. nezarae* and *P. saccharalis* lived 27–47 (Figure 1) and 26–31 days (Figure 3), respectively. However, starved individuals survived for only 3–5 days in both wasp species (Figure 1 and Figure 3). Populations of *O. nezarae* and *P. saccharalis* found in Asia exhibited survival periods under starvation conditions that were somewhat similar to our observations, with lifespans of two days [33] and four days [34], respectively. Females exhibited either similar lifespans to males or outlived them when food resources like water and honey were absent. The ability of females to live longer without resources might be attributed to host-feeding behavior like those reported by Takasu and Hirose [35], where adult *O. nezarae* females feed on host fluids before egg deposition. The increased longevity of females could also be associated with their larger body size [36]. Consistent with this, honey-deprived, oviposition host-provided females survived much longer than males (Figure 1 and Figure 3). However, honey supply suppressed this longevity advantage, indicating that males rely more on alternative sugar sources such as extrafloral nectar and hemipteran honeydew [17]. These results suggest that females have an advantage over males in the field during water and sugar shortages because they are less sensitive to the scarcity of these resources and can meet some of their nutritional needs through host feeding [37,38,39].

#### 4.1.2. Host Parasitism

According to our data, the parasitism rate of *M. cribraria* eggs by female *O. nezarae* varied with diet (Figure 2A). A honey supply led to the highest parasitism regardless of water availability, showing that sugar plays an important role in this species’ total host–parasitism capacity over the adult lifespan. Interestingly, this sugar effect was not observed for *P. saccharalis* females (Figure 4C). We found a decline in parasitism with age in both wasp species, which is suggestive of a synovigenic strategy, as it implies that females exhaust their nutritional reserves as they age and increasingly rely on sugars to compensate for this nutritional loss [14]. However, additional studies that include a comprehensive examination of ovaries and egg maturation rates using the ovigeny index would be necessary to determine the ovigeny status of *O. nezarae* and *P. saccharalis*.

#### 4.1.3. Wasp Emergence

Our study found that the reproductive capacity of *O. nezarae* and *P. saccharalis* was influenced by different factors. For both species, the highest offspring emergence was observed for young females (Figure 2C and Figure 4C), suggesting that age is also an important factor. For *O. nezarae*, offspring emergence depended on the diet and age of the female, with honey being particularly influential (Figure 2C). This is consistent with synovigenic species, such as *Pachycrepoideus vindemmiae*, that require additional nutrition to sustain oogenesis [17,36]. Conversely, the reproductive ability of *P. saccharalis* was influenced by age rather than diet, as honey-fed and honey-deprived females showed no significant differences in offspring production (Table 2), suggesting that *P. saccharalis* might be pro-ovigenic.

#### 4.1.4. Offspring Sex Ratio

The sex ratios of offspring in parasitic wasps are influenced by the age and diet of the mother [40], and our results provide more evidence that the offspring sex ratio of *O. nezarae* and *P. saccharalis* is related to maternal age and diet. The proportion of female offspring continually decreased as females aged. The decline was more substantial in honey-deprived wasps than in their honey-fed counterparts (Figure 2D and Figure 4D). The females used in this study mated following their emergence, but after 24 h, they were separated. As females had no opportunity to remate, they could not replenish their sperm storage during egg-laying. These results suggest that females of both parasitoids might require multiple mating during their lifetime to sustain the production of female offspring.

### 4.2. Effect of Host-to-Parasitoid Ratios and Host Exposure Times on the Biology of O. nezarae and P. Saccharalis

The number of host eggs and the time available to parasitize these eggs also affect their offspring emergence and sex ratio. Stable parasitism (Figure 5A and Figure 6A) and maximum yield of parasitoid offspring (Figure 5C and Figure 6C) in *O. nezarae* occurred when a single or a group of female wasps was exposed to a low host density. The exposure time required to maximize parasitism depended upon the number of parasitoids. The highest parasitism, with a female:male sex ratio of at least ~1:1, was achieved with a three- and five-day exposure duration for a single wasp or a one-day exposure time for a group of wasps. For *P. saccharalis*, the proportion of host eggs parasitized (Figure 7A and Figure 8A) and emerged offspring (Figure 7C and Figure 8C) reached a plateau at a density of 48 host eggs over three days of exposure time when hosts were provided to both single or groups of *P. saccharalis* females. In this study, host density affected both *O. nezarae* and *P. saccharalis* parasitism and reproductive performance differently depending on the exposure time. In a study using a single *O. nezarae* or *P. saccharalis* females, parasitism decreased as the host-to-parasitoid ratio increased across all host exposure times (Figure 5A and Figure 7A). However, when *O. nezarae* females were released in a group, parasitism was the lowest at one day of exposure and the highest at three and five days. Similarly, host parasitism was the highest at three and five days of exposure and the lowest at one day when *O. nezarae* females were released singly. This suggests that a one-day exposure time might not be enough time for *O. nezarae* females to parasitize large host patches to suppress host nymph populations. A previous study found that host exposure time significantly impacted the parasitism rate of *Telenomus remus* Nixon on *Spodoptera litura* Fabricius eggs, with the highest parasitism rate observed after exposure for one day [41]. In this study, longer exposure times increased egg parasitism. However, when a single or a group of *P. saccharalis* females was exposed to host eggs, the level of egg parasitism after a one-day exposure was almost the same as that for five days. It indicates that a one-day exposure time might be enough for *P. saccharalis* females to parasitize up to a proportion of 0.90 available hosts in one day. Similarly, Wang and Shelton [42] did not find a major difference between host parasitism at two and three days of exposure using *Ostrinia nubilalis* Hübner (Lepidoptera: Pyralidae) eggs as host for *Trichogramma ostriniae* Peng and Chen (Hymenoptera: Trichogrammatidae). The possible explanation for the parasitism results is that *P. saccharalis* (specialist species) has better host-handling strategies (an average of 10.48 min) due to specificity [34] than *O. nezarae* (an average of 19.32 min) [35].

There was a major reduction in the offspring of *O. nezarae* or *P. saccharalis* at the lower densities of *M. cribraria* eggs exposed to a group of female parasitoids for five days. Longer exposure times (5 days) may result in superparasitism due to a shortage of unparasitized hosts, decreasing the successful development and emergence of parasitoid offspring [43]. Intraspecific competition for limited nutrients can also decrease wasp emergence. The nutrients within host eggs are usually enough for only one wasp larva to complete its development. The competition for limited nutrients causes cannibalism and increases parasitoid larval mortality [43]. A previous study found a negative correlation between the quantity of emerged adults and the host egg to parasitoid ratio at prolonged exposure [41].

For *O. nezarae*, the proportion of female offspring followed an inverse relationship between emerged wasps and sex ratio (female/total offspring) in single (Figure 5D) or multiple (Figure 6D) wasp release experiments. When the host egg: parasitoid ratio increased in the five-day exposure treatment, the proportion of female offspring increased in single-wasp release experiments (Figure 5D) but decreased in group wasp release experiments (Figure 6D). For *P. saccharalis*, the proportion of female offspring (Figure 7D and Figure 8D) followed a similar pattern with the emergence of wasps in both single and multiple-wasp release experiments. The lowest proportion of female offspring in both wasp species was observed at a 21:7 host egg: parasitoid ratio for prolonged exposure, possibly due to superparasitism, although our study did not provide direct evidence to confirm this phenomenon. This finding is aligned with Chen et al. [41], where a low host egg-to-parasitoid ratio (80:80) led to more male offspring in *T. remus*, with confirmed superparasitism.

## 5. Conclusions

The present study showed that consistent access to water and honey increased the survivability, parasitism capacity, and offspring sex ratios for two egg parasitoids. It was also demonstrated that a relatively low host-to-parasitoid ratio and host exposure time optimize the reproductive potential of *O. nezarae* and *P. saccharalis*. Both parasitic wasps are long-lived and can significantly contribute toward *M. cribraria* biocontrol. These findings call for additional research, such as the identification of flowering plants that can serve as nectar sources for *O. nezarae* and *P. saccharalis* in the field. Such research could contribute to the preservation and effectiveness of these biological control agents of *M. cribraria*.

## Figures and Tables

**Figure 1 insects-14-00755-f001:**
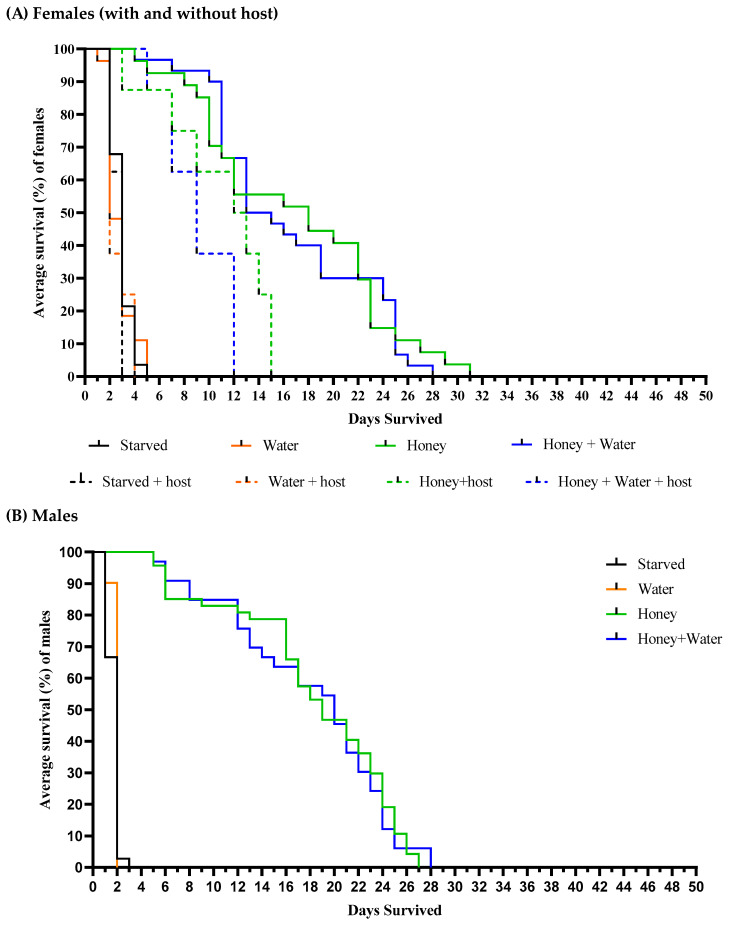
Effects of the constant supply of water, honey, and/or eggs of *Megacopta cribraria* on the survival curves of adult (**A**) female and (**B**) male adults of *Ooencyrtus nezarae*. Curves were estimated using the Kaplan–Meier method. The replicates individual wasps, respectively, for water, honey, water + honey, and starved (no water, no honey) were 30, 35, 59, and 30 for non-host provided females (**A**), 10, 10, 10, and 10 for host-provided females (**A**), and 41, 47, 33, and 36 for males (**B**).

**Figure 2 insects-14-00755-f002:**
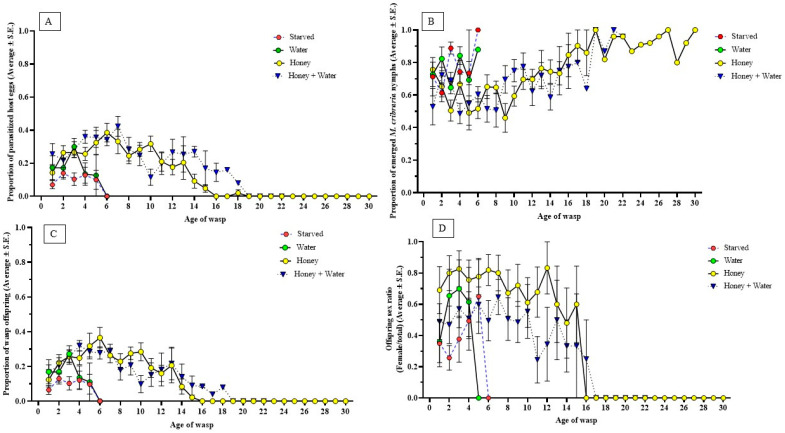
Effects of a constant supply of water and/or honey on life history traits of mated female *Ooencyrtus nezarae* reared with *Megacopta cribraria* eggs. Data show the (**A**) proportion of parasitized host eggs (Average ± S.E.), (**B**) proportion of emerged *M. cribraria* nymphs (Average ± S.E.), (**C**) proportion of wasp offspring (Average ± S.E.), and (**D**) the offspring sex ratio (female/total) (Average ± S.E.). Graphs were prepared using GraphPad Prism. (Sample size, *n* = 10).

**Figure 3 insects-14-00755-f003:**
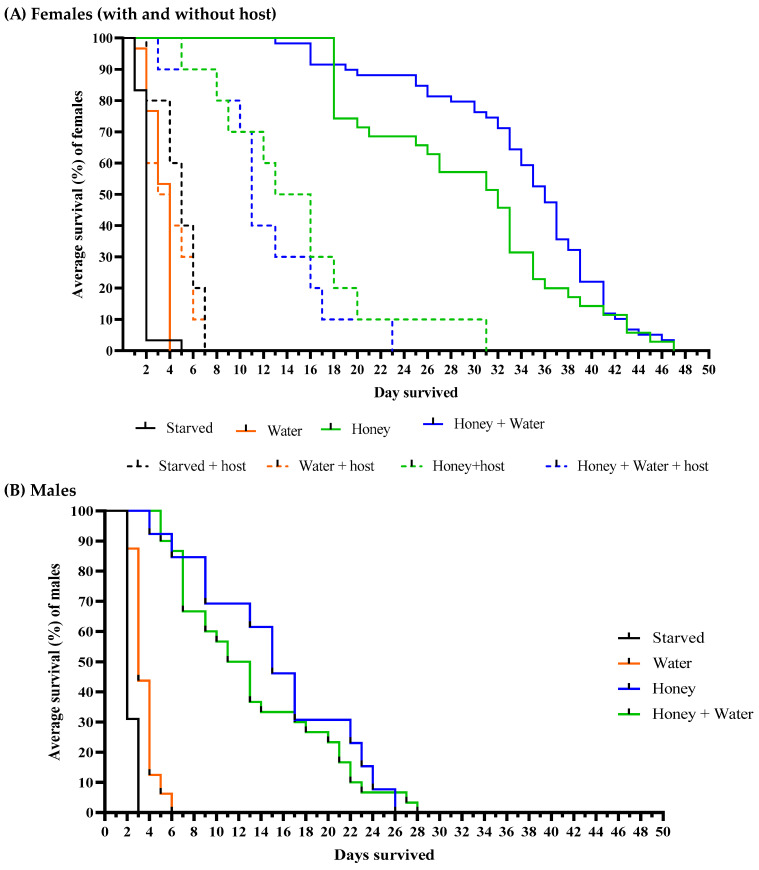
Effects of the constant supply of water, honey, and/or eggs of *Megacopta cribraria* on the survival curves of mated adult (**A**) female and (**B**) male *Paratelenomus saccharalis*. Curves were estimated using the Kaplan–Meier method. The numbers of replicates, respectively, for water, honey, water + honey, and starved (no water, no honey) were 27, 27, 30, and 28 for non-host provided females (**A**), 8, 8, 8, and 8 for host-provided females (**A**), and 16, 13, 30, and 29 for males (**B**).

**Figure 4 insects-14-00755-f004:**
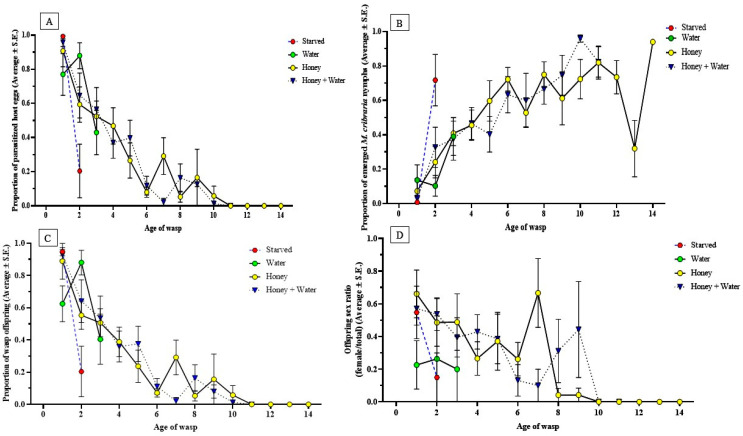
Effects of supply of water and/or honey on life-history traits of mated female *Paratelenomus saccharalis* reared on *Megacopta cribraria* eggs. Data show the (**A**) proportion of parasitized host eggs (Average ± S.E.), (**B**) proportion of emerged *M. cribraria* nymphs (Average ± S.E.), (**C**) proportion of wasp offspring (Average ± S.E.), and (**D**) the offspring sex ratio (female/total) (Average ± S.E.). Graphs were prepared using GraphPad Prism. (Sample size, *n* = 8).

**Figure 5 insects-14-00755-f005:**
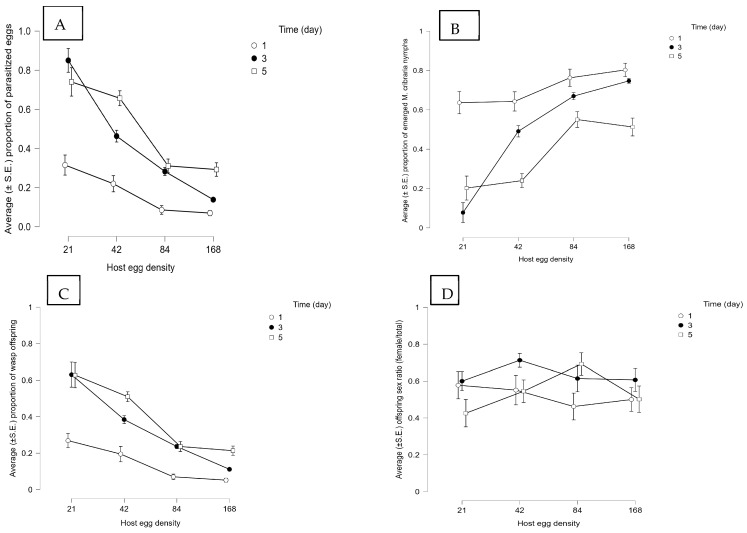
Effect of the host egg to parasitoid ratio and host egg exposure time on the performance and reproductive outcomes in single *Ooencyrtus nezarae* female release [sample size (*n*) = 20]. The presented results are the average (±S.E.) of the (**A**) proportion of parasitized *Megacopta cribraria* eggs, (**B**) proportion of emerged *M. cribraria* nymphs, (**C**) proportion of wasp offspring, and (**D**) offspring sex ratio (female/total).

**Figure 6 insects-14-00755-f006:**
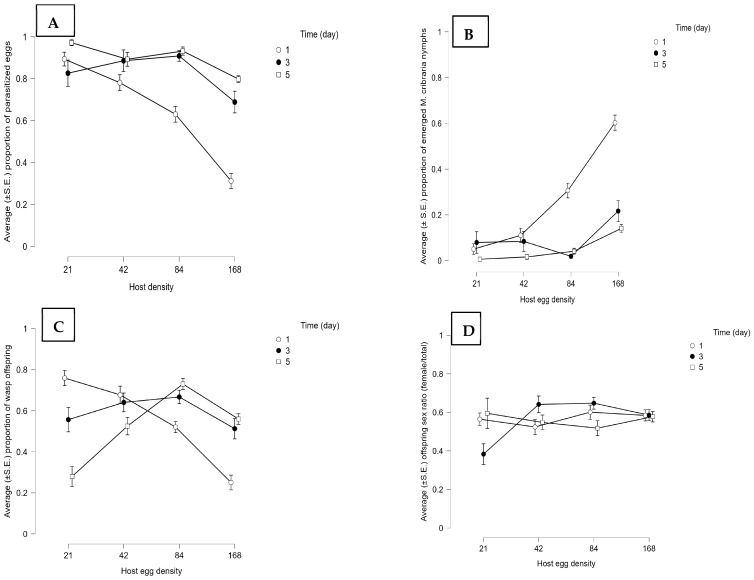
Effect of the host egg to parasitoid ratio and host egg exposure time on the performance and reproductive outcomes in group releases of seven *Ooencyrtus nezarae* females [sample size (*n*) = 20 groups]. The presented results are average (±S.E.) (**A**) proportion of parasitized *Megacopta cribraria* eggs, (**B**) proportion of emerged *M. cribraria* nymphs, (**C**) proportion of wasp offspring, and (**D**) offspring sex ratio (female/total).

**Figure 7 insects-14-00755-f007:**
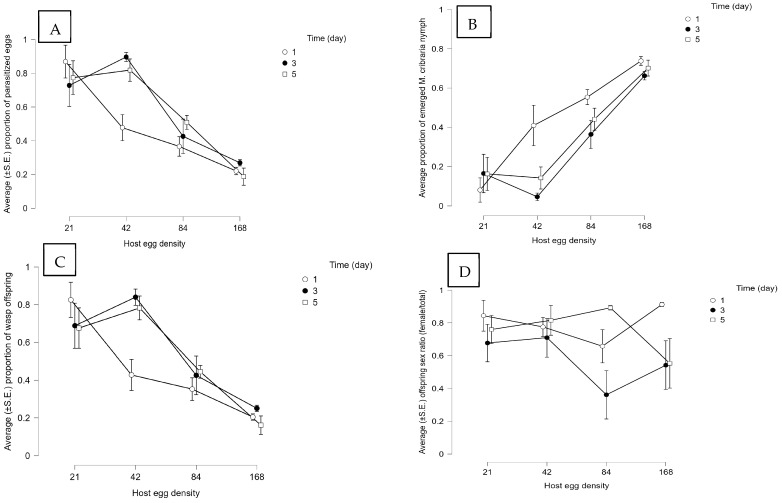
Effect of the host egg to parasitoid ratio and host egg exposure time on performance and reproductive outcomes in single *Paratelenomus saccharalis* female release [sample size (*n*) = 10]. The presented results are the average (±S.E.) (**A**) proportion of parasitized *M. cribraria* eggs, (**B**) proportion of emerged *M. cribraria* nymphs, (**C**) proportion of wasp offspring, and (**D**) offspring sex ratio (female/total).

**Figure 8 insects-14-00755-f008:**
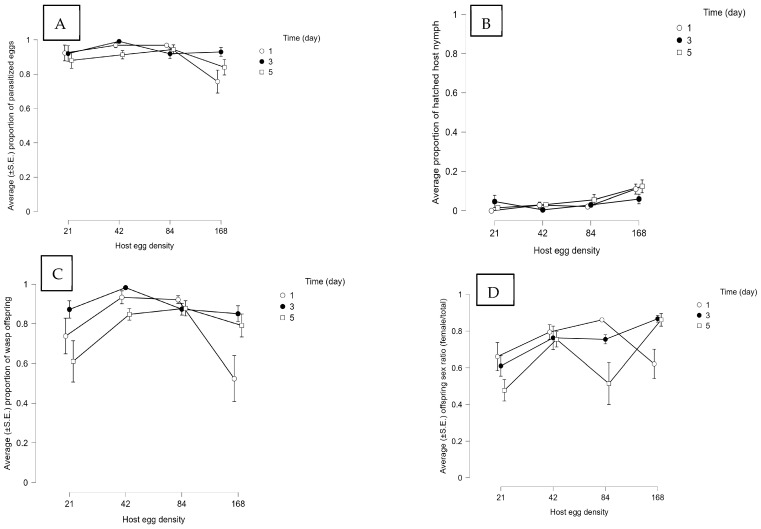
Effect of the host egg to parasitoid ratio and host egg exposure time on the performance and reproductive outcomes in group releases of seven *Paratelenomus saccharalis* females (sample size (*n*) = 10). The presented results are average (±S.E.) (**A**) proportion of parasitized *M. cribraria* eggs, (**B**) proportion of emerged *M. cribraria* nymphs, (**C**) proportion of wasp offspring, and (**D**) offspring sex ratio (female/total).

**Table 1 insects-14-00755-t001:** The longevity (average ± S.E) of *Ooencyrtus nezarae* and *Paratelenomus saccharalis* males and females with or without a host (*Megacopta cribraria* eggs) that were either starved or provided access to a constant supply of water, honey, or honey + water.

		Males			Females			Females + Host Eggs		
Parasitoid Species	Diet	Average ± S.E.	Range	Sample Size	Average ± S.E.	Range	Sample Size	Average ± S.E.	Range	Sample Size
^1^ *Ooencyrtus nezarae*	Starved	1.69 ± 0.79 b	1–3	36	1.93 ± 1.22 c	1–5	30	4.80 ± 1.50 b	2–7	10
	Water	1.90 ± 0.74 b	1–2	41	3.26 ± 1.22 c	1–4	30	3.90 ± 1.50 b	2–7	10
	Honey	18.27 ± 0.69 a	5–27	47	29.60 ± 1.13 b	18–47	35	14.80 ±1.50 a	5–31	10
	Honey + Water	17.93 ± 0.83 a	5–28	33	34.06 ± 0.87 a	13–47	59	12.30 ±0.83 a	3–23	10
	One-way ANOVA values	df = 3, 153; F = 152.09; *p* < 0.0001			df = 3, 153; F = 244.36; *p* < 0.0001			df = 3, 36; F = 12.90; *p* < 0.0001		
^2^ *Paratelenomus saccharalis*	Starved	2.31 ± 0.91 b	2–3	29	2.92 ± 0.96 b	2–5	28	2.62 ± 0.91 b	2–3	8
	Water	3.50 ± 1.22 b	2–6	16	2.74 ± 0.98 b	1–5	27	2.62 ± 0.91 b	2–4	8
	Honey	15.38 ± 1.36 a	4–26	13	16.85 ± 0.98 a	4–31	27	11.00 ± 0.91 a	3–15	8
	Honey + Water	13.26 ± 0.89 a	5–28	30	16.50 ± 0.93 a	4–28	30	9.12 ± 0.91 a	5–12	8
	One-way ANOVA values	df = 3, 84; F = 38.68; *p* < 0.0001			df = 3, 108; F = 68.23; *p* < 0.0001			df = 3, 28; F = 22.59; *p* < 0.0001		

^1^ For *Ooencyrtus nezarae*: Different letters in the same category indicate significant differences per Tukey’s test (*p* ≤ 0.05). ^2^ For *Paratelenomus saccharalis*: Different letters in the same category indicate significant differences per Tukey’s test (*p* ≤ 0.05).

**Table 2 insects-14-00755-t002:** Lifetime offspring production (Average ± S.E) of *Ooencyrtus nezarae* and *Paratelenomus saccharalis* females reared with *Megacopta cribraria* eggs that were either starved (no water, no honey) or fed water, honey, or honey + water.

Parasitoid Species	Diet	Wasp Offspring			Offspring Sex Ratio (Female/Total)		
		Average ± S.E.	Range	Sample Size	Average ± S.E.	Range	Sample Size
^1^ *Ooencyrtus nezarae*	Starved	8.70 ± 7.27 b	0–24	10	0.49 ± 0.10 ab	0–11	10
	Water	15.70 ± 7.27 b	0–68	10	0.32 ± 0.10 b	0–24	10
	Honey	63.00 ± 7.27 a	28–115	10	0.80 ± 0.10 a	21–108	10
	Honey + Water	55.40 ± 7.27 a	0–109	10	0.51 ± 0.10 a	0–70	10
^2^ *Paratelenomus saccharalis*	Starved	21.50 ± 5.98 b	17–24	8	0.50 ± 0.12 ns	0–22	8
	Water	21.75 ± 5.98 b	43–50	8	0.22 ± 0.12 ns	0–31	8
	Honey	63.62 ± 5.98 a	34–89	8	0.59 ± 0.12 ns	6–59	8
	Honey + Water	65.50 ± 5.98 a	32–92	8	0.54 ± 0.12 ns	1–54	8

^1^ For *Ooencyrtus nezarae*: Different letters in the same category indicate significant differences per Tukey’s test (*p* ≤ 0.05). ^2^ For *Paratelenomus saccharalis*: Different letters in the same category indicate significant differences per Tukey’s test (*p* ≤ 0.05). ns = not significant effect.

## Data Availability

The data presented in this study are available in this article.

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
