# Peer review of "Effects of Food Source Availability, Host Egg:Parasitoid Ratios, and Host Exposure Times on the Developmental Biology of Megacopta cribraria Egg Parasitoids"

_insects, 2023, doi:10.3390/insects14090755_

Round 1
Reviewer 1 Report
I have read with attention the ms titled: "Developmental Biology of Megacopta Cribraria Egg Parasitoids is Influenced by the Availability of Food Sources, Host Egg to Parasitoid Ratios, and Host Expoure Times ", and authored by Sanower Warsi, Ana M. Chicas-Mosier,Rammohan R. Balusu, Alana L. Jacobson and Henry Y. Fadamiro.
Below you will find my remarks, comments, and suggestions.
I suggest to Change the title in:
Effects of food source availability, host egg-parasitoid ratios, and host exposure times on the developmental biology of Megacopta cribraria egg parasitoids
Title and summary to be rewritten because they are not clear
A little in the past and a little in the present, equalize the tense of the verbs.
Line 34-36 the sentence is not clear and replace the colon after the word “host eggs:” with “/”
Line 37 change the sentence because it's not clear
38-40 unclear sentence, please rearranged.
Both the summary and the abstract are not clear and need to be rewritten.
Line 51 please change but with “hence” or “so”
Line 98 author should explain why they hypothesized that the presence of food would enhance parasitoid species’ emergence sincerely it is not clear.
Line 100 please delete low before host/parasitoid ratio
Line 124 please change “In” with “in”
Line 125 please put the reference at the end of the sentence.
Line 127 please change were with was it seems a singular form.
Line 140-141 please use italic for the name of insect moreover use the abbreviated form
Line 142 I suggest to use plurals as the verb
Line 162-165 this sentence is not clear please rearrange. Why humidity should determine insect desiccation?
Line 177-178 please use italic for the name of insect moreover use the abbreviated form, moreover, authors should use capital letters only at the beginning of sentences and for insect names throughout the manuscript, even in paragraph titles
line 181 please close the bracket
lines 181-183 please check I think Authors should add at least a comma or to rearrange the sentence.
Lines 190-192 this sentence is unclear and should be rearranged.
Lines 235-238 this sentence is unclear and must be rearranged.
Lines 250-253 please use italic just for the name of insect moreover use the abbreviated form, moreover, authors should use capital letters only at the beginning of sentences and for insect names throughout the manuscript, even in paragraph titles
Line 256-259 There is an apparent incongruence because authors wrote: When O. nezarae females were provided with different diet regimens and placed in the presence of an oviposition host, their survival rate was significantly lower compared to when they were in the absence of an oviposition host (Figure 1A).
But in the table 1 the longevity of starved and water provided females with eggs had a longer longevity than females without hosts. Please check.
Line 257 please delete “an oviposition” and add “a”
Lines 278-279 please rearrange the sentence; I suggest to simplify in “with or without hosts”
Table 1. Because the number of repetitions is different it is preferable to use the post-hoc Bonferroni test instead of the Tukey test.
Please apply that.
Lines 291-292 This sentence is not clear what do the authors mean by prevalence? Please rearrange the sentence.
Lines 292-293 Maybe “A steeper decline was observed in the water and starved diet treatments compared to the others” could be more understandable respect what author wrote. I suggest to rephrase the sentence,
lines 298 -306 This part should be rearranged because it could be described in a simpler way without many words.
For example, “The presence of parasitoids influences the % of host hatched eggs because...
The data should be presented by the authors only once; however, they are often reported in the figures or tables and in the text, the authors should avoid these unnecessary repetitions.
Lines 307-320 This part is really difficult to read and many of the data are repetitions of data already present in the figures. The authors must rearrange it.
Figure 2 Please check the what is written on the ordinates because spaces are missing and the symbol ± must be replaced.
Line 334. Legend of table 2. Please see above what I wrote for the previous legend.
Line 339-340 Please use the abbreviate name of insect
Line 343 344 this sentence should be moved to discussion .
Line 364-367 This part is not clear, this seems inconsistent if the results are not statistically significant, the numerical differences can be due to the case for which they are not to be considered different.
The authors must rewrite this part.
Lines 364-400 Also in this part the authors repeat many of the results that are already presented as figures, this is absolutely to be avoided. All the results must be rearranged by taking into account this suggestion
Lines 405-409 Please use the abbreviate name of insect
Lines 420-422 may be “presence” could be better than “release”.
Table 3 The authors should explain the meaning of Z in the legend
Table 3 and 4 should be placed as additional material and therefore removed from the manuscript.
Line 430 Authors wrote “Significant values (p < 0.05) are in bold font” but also values = 0.05 are in bold . Please change.
Lines 502.504 In this case it seems to me that the subject is populations so I believe that this sentence should be rearranged
.
Populations lived for. Please change.
Lines 5050-508 there is another apparent incongruence : Authors wrote: Females had similar or longer lifespans than males when food resources like water and honey were absent.
But then they wrote: that adult O. nezarae females feed on host fluids before egg deposition, which may help to explain why females can live longer than males when resources are absent. Please check and rearrange this sentence if they had a similar lifespan.
Moreover, what is the size of the individuals of the two sexes, are males smaller? In this case, the dimensions could also affect the longevity and this is a well -known phenomenon for insects, please evaluate this thing and add in the Some References. Larger individuals have more reserves…
Lines 516-517 this part must be re arranged because it is not clear.
Authors should use the word “ host feeding” to describe the behavior of the females . There are a lot pf papers on this kind of behavior
Line 491-523 That the presence of sugary substances or host feeding can influence both longevity and offspring and many other biological characteristics of parasitoids is a very well -known thing. The authors should mention more papers that deal with these topics.
Lines 516-523
The results discussed could be strongly conditioned by the ovigeny (synovigenic or pro ovigenic) of the two species of parasitoids, but the authors not only have not assessed it but do not even take it into consideration of the discussion. I highly recommend evaluating the ovigeny of the two species and rearrange this paragraph taking into account the ovigeny
Line 516-525
The results discussed in this paragraph can also be strongly influenced by ovigeny of the two parasitoids. Please rearrange this part.
Lines 527-529
I found another apparent incongruence, Authors wrote: The highest emergence of offspring was primarily observed from eggs laid by young female parasitoids (Figures 2C and 4C). This indicates that the nutritional status and age of both parasitoid species affect their ability to produce offspring.
I do not understand the link with the age of parasitoids. Moreover, also this biological characteristic could be affected by ovigeny index of parasitoids.
The study has been well-designed. In terms of the statistical analysis conducted, I suggest replacing the Tukey test with a more conservative post hoc test, such as the Bonferroni. This change is recommended because the number of repetitions between the different theses often varies. By adopting a more conservative approach, some of the differences that are currently deemed significant may no longer hold true.
Although I'm not a native speaker, I find the text somewhat challenging to comprehend at times. Simplifying and clarifying certain concepts could enhance its effectiveness. Notably, it appears that the authors have limited familiarity with the historical literature relevant to the topics under investigation.
The authors have not investigated the ovigeny of the two parasitoids, which could potentially impact many of their results. Please study the following paper: Jervis, M. A., Heimpel, G. E., Ferns, P. N., Harvey, J. A., Neil, A. C., & Ecology, A. (2001). Life-History Strategies in Parasitoid Wasps : A Comparative Analysis of ’ Ovigeny. Jpurnal of Animal Ecology, 70(3), 442–458.
Additionally, the phenomenon of host feeding is discussed without utilizing the specific terminology associated with it.
Furthermore, the presentation of results is quite intricate, partly due to the dual format of data presentation (tables or figures and text). It would be advisable to avoid this dual approach.
Given these concerns, I recommend a thorough examination of ovigeny and a complete revision of the results section
I therefore believe that the manuscript must undergo a major revision before deserving publication in Insects.
I have read with attention the ms titled: "Developmental Biology of Megacopta Cribraria Egg Parasitoids is Influenced by the Availability of Food Sources, Host Egg to Parasitoid Ratios, and Host Expoure Time ", and authored by Sanower Warsi, Ana M. Chicas-Mosier,Rammohan R. Balusu, Alana L. Jacobson and Henry Y. Fadamiro.
Below you will find my remarks, comments, and suggestions.
I suggest to Change the title in:
Effects of food source availability, host egg-parasitoid ratios, and host exposure times on the developmental biology of Megacopta cribraria egg parasitoids
Title and summary to be rewritten because they are not clear
A little in the past and a little in the present, equalize the tense of the verbs.
Line 34-36 the sentence is not clear and replace the colon after the word “host eggs:” with “/”
Line 37 change the sentence because it's not clear
38-40 unclear sentence, please rearranged.
Both the summary and the abstract are not clear and need to be rewritten.
Line 51 please change but with “hence” or “so”
Line 98 author should explain why they hypothesized that the presence of food would enhance parasitoid species’ emergence sincerely it is not clear.
Line 100 please delete low before host/parasitoid ratio
Line 124 please change “In” with “in”
Line 125 please put the reference at the end of the sentence.
Line 127 please change were with was it seems a singular form.
Line 140-141 please use italic for the name of insect moreover use the abbreviated form
Line 142 I suggest to use plurals as the verb
Line 162-165 this sentence is not clear please rearrange. Why humidity should determine insect desiccation?
Line 177-178 please use italic for the name of insect moreover use the abbreviated form, moreover, authors should use capital letters only at the beginning of sentences and for insect names throughout the manuscript, even in paragraph titles
line 181 please close the bracket
lines 181-183 please check I think Authors should add at least a comma or to rearrange the sentence.
Lines 190-192 this sentence is unclear and should be rearranged.
Lines 235-238 this sentence is unclear and must be rearranged.
Lines 250-253 please use italic just for the name of insect moreover use the abbreviated form, moreover, authors should use capital letters only at the beginning of sentences and for insect names throughout the manuscript, even in paragraph titles
Line 256-259 There is an apparent incongruence because authors wrote: When O. nezarae females were provided with different diet regimens and placed in the presence of an oviposition host, their survival rate was significantly lower compared to when they were in the absence of an oviposition host (Figure 1A).
But in the table 1 the longevity of starved and water provided females with eggs had a longer longevity than females without hosts. Please check.
Line 257 please delete “an oviposition” and add “a”
Lines 278-279 please rearrange the sentence; I suggest to simplify in “with or without hosts”
Table 1. Because the number of repetitions is different it is preferable to use the post-hoc Bonferroni test instead of the Tukey test.
Please apply that.
Lines 291-292 This sentence is not clear what do the authors mean by prevalence? Please rearrange the sentence.
Lines 292-293 Maybe “A steeper decline was observed in the water and starved diet treatments compared to the others” could be more understandable respect what author wrote. I suggest to rephrase the sentence,
lines 298 -306 This part should be rearranged because it could be described in a simpler way without many words.
For example, “The presence of parasitoids influences the % of host hatched eggs because...
The data should be presented by the authors only once; however, they are often reported in the figures or tables and in the text, the authors should avoid these unnecessary repetitions.
Lines 307-320 This part is really difficult to read and many of the data are repetitions of data already present in the figures. The authors must rearrange it.
Figure 2 Please check the what is written on the ordinates because spaces are missing and the symbol ± must be replaced.
Line 334. Legend of table 2. Please see above what I wrote for the previous legend.
Line 339-340 Please use the abbreviate name of insect
Line 343 344 this sentence should be moved to discussion .
Line 364-367 This part is not clear, this seems inconsistent if the results are not statistically significant, the numerical differences can be due to the case for which they are not to be considered different.
The authors must rewrite this part.
Lines 364-400 Also in this part the authors repeat many of the results that are already presented as figures, this is absolutely to be avoided. All the results must be rearranged by taking into account this suggestion
Lines 405-409 Please use the abbreviate name of insect
Lines 420-422 may be “presence” could be better than “release”.
Table 3 The authors should explain the meaning of Z in the legend
Table 3 and 4 should be placed as additional material and therefore removed from the manuscript.
Line 430 Authors wrote “Significant values (p < 0.05) are in bold font” but also values = 0.05 are in bold . Please change.
Lines 502.504 In this case it seems to me that the subject is populations so I believe that this sentence should be rearranged
.
Populations lived for. Please change.
Lines 5050-508 there is another apparent incongruence : Authors wrote: Females had similar or longer lifespans than males when food resources like water and honey were absent.
But then they wrote: that adult O. nezarae females feed on host fluids before egg deposition, which may help to explain why females can live longer than males when resources are absent. Please check and rearrange this sentence if they had a similar lifespan.
Moreover, what is the size of the individuals of the two sexes, are males smaller? In this case, the dimensions could also affect the longevity and this is a well -known phenomenon for insects, please evaluate this thing and add in the Some References. larger individuals have more reserves…
Lines 516-517 this part must be re arranged because it is not clear.
Authors should use the word “ host feeding” to describe the behavior of the females . There are a lot pf papers on this kind of behavior
Line 491-523 That the presence of sugary substances or host feeding can influence both longevity and offspring and many other biological characteristics of parasitoids is a very well -known thing. The authors should mention more papers that deal with these topics.
Lines 516-523
The results discussed could be strongly conditioned by the ovigeny (synovigenic or pro ovigenic) of the two species of parasitoids, but the authors not only have not assessed it but do not even take it into consideration of the discussion. I highly recommend evaluating the ovigeny of the two species and rearrange this paragraph taking into account the ovigeny
Line 516-525
The results discussed in this paragraph can also be strongly influenced by ovigeny of the two parasitoids. Please rearrange this part.
Lines 527-529
I found another apparent incongruence, Authors wrote: The highest emergence of offspring was primarily observed from eggs laid by young female parasitoids (Figures 2C and 4C). This indicates that the nutritional status and age of both parasitoid species affect their ability to produce offspring.
I do not understand the link with the age of parasitoids. Moreover, also this biological characteristic could be affected by ovigeny index of parasitoids.
The study has been well-designed. In terms of the statistical analysis conducted, I suggest replacing the Tukey test with a more conservative post hoc test, such as the Bonferroni. This change is recommended because the number of repetitions between the different theses often varies. By adopting a more conservative approach, some of the differences that are currently deemed significant may no longer hold true.
Although I'm not a native speaker, I find the text somewhat challenging to comprehend at times. Simplifying and clarifying certain concepts could enhance its effectiveness. Notably, it appears that the authors have limited familiarity with the historical literature relevant to the topics under investigation.
The authors have not investigated the ovigeny of the two parasitoids, which could potentially impact many of their results. Please study the following paper: Jervis, M. A., Heimpel, G. E., Ferns, P. N., Harvey, J. A., Neil, A. C., & Ecology, A. (2001). Life-History Strategies in Parasitoid Wasps : A Comparative Analysis of ’ Ovigeny. Jpurnal of Animal Ecology, 70(3), 442–458.
Additionally, the phenomenon of host feeding is discussed without utilizing the specific terminology associated with it.
Furthermore, the presentation of results is quite intricate, partly due to the dual format of data presentation (tables or figures and text). It would be advisable to avoid this dual approach.
Given these concerns, I recommend a thorough examination of ovigeny and a complete revision of the results section
I therefore believe that the manuscript must undergo a major revision before deserving publication in Insects.
Author Response
Please see the attached response file. Thank you

Reviewer 2 Report
The authors conducted several laboratory experiments aimed to determine optimal conditions (adult food source, host:parasitiod ratio, and host exposure time) for the highest production of adults females in mass rearing of Paratelenomus saccharalis and Ooencyrtus nezarae, egg parasitoids of the kudzu bug, Megacopta cribraria. The experiments were well planned and conducted; the data were correctly analyzed and mostly properly presented; conclusions are supported by the data; the obtained results can be used for the optimization of laboratory and mass rearing of biocontrol agents. Thus, the manuscript can be published, although a number of important corrections and improvements should be made before publication (see below). Generally, the manuscript was not very well prepared. There are a lot of small mistakes that hampered the reading and understanding of the paper. Possibly, I have missed some of them. Therefore the whole manuscript (particularly the Results section) should be carefully checked by the authors.
Line100: “when low host: parasitoid ratio is low” – the first ‘low” should be deleted.
Line 124: “In” should not start with a capital letter.
Line 140: Paratelonomus saccharalis and Ooencyrtus nezarae should be in Italics font.
Lines 180-181: The second bracket is absent in this sentence.
Line 203: Please, explain what you mean “un-ascribed eggs”.
Lines 212-219: In this sentence, two-way ANOVA is mentioned but only one factor (food source treatment) is indicated. Besides, as far as I know, the Tukey’HSD test follows not two-way but one-way ANOVA. Finally, the results of the Tukey’s test are not shown in these figures (see also comments to lines 332 and 403). Please, check everything and explain your statistical methods in more details.
Lines 227-235: First, the results of the statistical analysis (significances of pairwise differences) are given NOT in the Figures 5-8 (that are listed in this sentence) BUT in Tables 3 and 4 (that are not mentioned in this sentence). Second, although this sentence is started with “The interactive effects...” the interaction of the two factors (host: parasitoid ratio and host exposure time) was NOT statistically evaluated in these tables.
Line 252: Ooencyrtus nezarae should be in Italics font.
Lines 265-267: The meanings of these two sentences are almost the same. Possibly, the first one can be deleted?
Lines 280 and 337: Please, indicate clearly whether you mean a column (data for two species) or a half of the column (data for one species).
Lines 285, 304, 308 etc.: Unfortunately, I can’ find Tables S1 and S2 (see also comments to lines 608 and 612). Possibly, these tables were lost during the submission process?
Line 311: It is seen from Table 2 that honey-fed females of Ooencyrtus nezarae produced an average of 63 offspring whereas females with access to only water produced 15.7 offspring. Thus, the difference is NOT “twice as high” but almost 4 times higher. “Twice as high” is the difference between females with and without access to water.
Lines 324 and 326: Figure 3 includes only two graphs (A and B) showing survivorship curves for males and females of P. saccharalis. The Ooencyrtus nezarae offspring sex ratio is shown NOT in Figure 3D but in Figure 2D.
Lines 331 and 402: It is clearly seen that NOT the average ± SE but the average + SE (only the upper part of SE bar) is shown in these figures.
Line 332: I can’t find any result of the Tukey’s HSD test (which estimates the significance of pairwise differences) in this figure. Besides, this test usually follows not two-way but one-way ANOVA and, moreover, there is only one factor (feeding regime) in this figure.
Line 339: Paratelenomus saccharalis should be in Italics font.
Line 403: I can’t find any result of the Tukey’s HSD test (which estimates the significance of pairwise differences) in this figure.
Line 407: Ooencyrtus nezarae should be in Italics font.
Lines 432 and 437: NOT “the host egg to parasitoid ratio” but only the number of hosts is shown in these figures (the number of parasitoids is NOT shown). Therefore the difference between the experimental treatments shown in figures 5 and 6 (that is the number of female wasps) is not clear.
Lines 460 and 464: NOT “the host egg to parasitoid ratio” but only the number of hosts is shown in these figures (the number of parasitoids is NOT shown). Therefore the difference between the experimental treatments shown in figures 7 and 8 (that is the number of female wasps) is not clear.
Lines 609 and 612: Tables S1 and S2 should be somehow included in the manuscript or the actual link should be indicated. Otherwise, it is not possible to properly evaluate the results of the study.
Author Response

(The authors gave the same response as above.)

Round 2
Reviewer 1 Report
I am pleased to see that all the suggestions have been taken into account by the authors hence I think the ms deserves publication.
I am pleased to see that all the suggestions have been taken into account by the authors hence I think the ms deserves publication.
Reviewer 2 Report
The authors have considered al my comments. I have only one more correction to the new title (lines 2-3) that can be made in the proofs: “host egg-parasitoid ratios” should be replaced by “host egg:parasitoid ratios” (replace dash ‘-’ with colon ‘:’).